# Cataract Surgery with Intraocular Lens Implantation in Juvenile Idiopathic Arthritis-Associated Uveitis: Outcomes in the Era of Biological Therapy

**DOI:** 10.3390/jcm10112437

**Published:** 2021-05-31

**Authors:** Elena Bolletta, Marco Coassin, Danilo Iannetta, Valentina Mastrofilippo, Raffaella Aldigeri, Alessandro Invernizzi, Luca de Simone, Fabrizio Gozzi, Alessandro De Fanti, Michela Cappella, Chantal Adani, Alberto Neri, Antonio Moramarco, Michele De Maria, Carlo Salvarani, Luigi Fontana, Luca Cimino

**Affiliations:** 1Ocular Immunology Unit, Azienda USL-IRCCS di Reggio Emilia, 42121 Reggio Emilia, Italy; elena.bolletta@ausl.re.it (E.B.); valentina.mastrofilippo@ausl.re.it (V.M.); luca.desimone@ausl.re.it (L.d.S.); fabrizio.gozzi@ausl.re.it (F.G.); cahntal.adani@ausl.re.it (C.A.); 2Ophthalmology, University Campus Bio-Medico, 00128 Rome, Italy; m.coassin@unicampus.it; 3Eye Unit, Azienda USL-IRCCS di Reggio Emilia, 42121 Reggio Emilia, Italy; danilo.iannetta@ausl.re.it (D.I.); alberto.neri@ausl.re.it (A.N.); antonio.moramarco@ausl.re.it (A.M.); michele.demaria@ausl.re.it (M.D.M.); luifonta@gmail.com (L.F.); 4Department of Medicine and Surgery, University of Parma, 43126 Parma, Italy; raldige7@unipr.it; 5Department of Biomedical and Clinical Science “Luigi Sacco”, Eye Clinic, Luigi Sacco Hospital, University of Milan, 20157 Milan, Italy; alessandro.invernizzi@gmail.com; 6Pediatric Rheumatology Unit, Azienda USL–IRCCS di Reggio Emilia, 42121 Reggio Emilia, Italy; alessandro.defanti@ausl.re.it (A.D.F.); michela.cappella@ausl.re.it (M.C.); 7Rheumatology Unit, Department of Internal Medicine, Azienda USL–IRCCS di Reggio Emilia, 42121 Reggio Emilia, Italy; carlo.salvarani@ausl.re.it

**Keywords:** uveitis, juvenile idiopathic arthritis, JIA, biologicals, cataract surgery, intraocular lens, IOL

## Abstract

This study compared the outcomes of cataract surgery with intraocular lens (IOL) implantation in patients with juvenile idiopathic arthritis (JIA)-associated chronic anterior uveitis treated with antimetabolite drugs and systemic corticosteroids (Non-Biological Group) versus patients treated with antimetabolites and biological drugs (Biological Group). A cohort of patients with cataract in JIA-associated uveitis undergoing phacoemulsification with IOL implantation was retrospectively evaluated. The main outcome was a change in corrected distance visual acuity (CDVA) in the two groups. Ocular and systemic complications were also recorded. The data were collected preoperatively and at 1, 12, and 48 months after surgery. Thirty-two eyes of 24 children were included: 10 eyes in the Non-Biological Group and 22 eyes in the Biological Group. The mean CDVA improved from 1.19 ± 0.72 logMAR preoperatively to 0.98 ± 0.97 logMAR at 48 months (*p* = 0.45) in the Non-Biological Group and from 1.55 ± 0.91 logMAR preoperatively to 0.57 ± 0.83 logMAR at 48 months (*p* = 0.001) in the Biological Group. The postoperative complications, including synechiae, cyclitic membrane, IOL explantation, glaucoma, and macular edema, were not statistically different between the two groups. An immunosuppressive treatment with biological drugs can improve the visual outcome after cataract surgery in patients with JIA-associated uveitis, but it does not significantly reduce postoperative ocular complications.

## 1. Introduction

Cataract is a frequent complication in juvenile idiopathic arthritis (JIA)-associated uveitis that occurs in 20–64% of children with JIA. It is caused by posterior synechiae, chronic inflammation, and corticosteroid treatment [1,2]. If surgery is not promptly performed, the formation of cataracts during visual development can ultimately lead to further complications, such as amblyopia and strabismus [3]. Cataract surgery in these patients is challenging, and intraocular lens (IOL) implantation is controversial because of postoperative complications such as anterior and posterior synechiae, pupillary membrane, and secondary posterior capsule opacification (PCO) [4]. The susceptibility to develop such complications is attributed to uncontrolled preoperative and surgically induced inflammation [5]. IOL implantation can further stimulate ocular inflammation by serving as a scaffold for inflammatory cells and debris, thus increasing the probability of cyclitic membrane formation with a subsequent hypotony and phthisis bulbi [1]. Other postoperative complications include secondary glaucoma, macular edema (ME), and retinal detachment [6].

Modern surgical techniques with improved microsurgery instruments, IOL design, and materials have resulted in an increased success rate in uveitic cataract surgery [7]. Perioperative immunosuppression also plays a key role. Previous studies have recommended an anterior chamber (AC) free from inflammatory cells for at least 3 months before cataract surgery and a well-controlled disease throughout the postoperative period [7,8,9]. This approach reduces the risk of postoperative ME and ocular damage from postoperative inflammation [10]. Immunosuppressive therapy improves the medical management of patients incompletely responsive to corticosteroids or with unacceptable steroid-induced side effects [7]. Methotrexate (MTX) is the most commonly prescribed immunosuppressor in pediatric uveitis, although 27–48% of patients do not achieve complete inflammation control with this agent [11]. In such patients, the advent of biological drugs has provided an additional tool for controlling inflammation refractory to conventional immunosuppressive therapy [12,13]. In particular, the randomized placebo-controlled trial (SYCAMORE Study) proved the association of adalimumab (ADA) with MTX to be effective in preventing treatment failure [14]. The concept of adequate immunosuppression, consisting of a zero tolerance for inflammation, improved the outcomes of cataract surgery in JIA-associated uveitis [7]. Several studies demonstrated good results from phacoemulsification with primary IOL implantation performed in JIA-associated uveitis when first-line immunosuppressors (antimetabolites) such as MTX or azathioprine (AZA) were used perioperatively [1,3,8,9,15,16,17,18,19,20,21,22].

The purpose of this study was to compare the outcomes of cataract surgery with IOL implantation in patients affected by JIA-associated chronic anterior uveitis treated with an antimetabolite drug (MTX or AZA) and systemic corticosteroids versus patients under a combined treatment with MTX or AZA and a biological drug.

## 2. Materials and Methods

This retrospective study included consecutive patients affected by cataract in JIA–associated chronic anterior uveitis who underwent cataract surgery between 2005 and 2018 at the Ocular Immunology Unit, Azienda Unità Sanitaria Locale (AUSL)—IRCCS di Reggio Emilia, Reggio Emilia, Italy.

The diagnosis of JIA was made by a rheumatologist, according to the American Rheumatism Association’s clinical diagnostic criteria, and to the classification criteria of the International League of Associations for Rheumatology (ILAR) [23].

Patients operated on between 2005 and 2011 were treated with a first-line immunosuppressive drug (MTX or AZA) and systemic corticosteroids (Non-Biological Group), while patients who underwent surgery between 2011 and 2018 were treated with an immunosuppressive agent (MTX or AZA) and a tumor necrosis factor alpha (TNFα) inhibitor, such as adalimumab (ADA), infliximab (IFX), or tocilizumab (TCZ) (Biological Group).

Data collection included corrected distance visual acuity (CDVA), intraocular pressure (IOP), orthoptic evaluation, slit lamp biomicroscopy, fundus examination, and optical coherence tomography (OCT).

Cataract surgery was performed under general anesthesia by two experienced ophthalmic surgeons. All patients underwent phacoemulsification with primary posterior chamber foldable hydrophobic acrylic IOL implantation in the capsular bag using two strategies: either intraoperative posterior capsulorhexis, anterior vitrectomy, and optic capture or basal iridectomy, followed by postoperative Nd:YAG laser posterior capsulotomy within one month after surgery. When band keratopathy was present, scrubbing of the corneal surface with a solution of ethylenediaminetetraacetic acid (EDTA) 0.35% was performed intraoperatively.

The primary outcome was the change in CDVA compared between the two groups. Intraoperative and postoperative ocular complications, including synechiae, pupillary membrane, secondary PCO, ME, epiretinal membrane, retinal detachment, and ocular hypertension or hypotony, were assessed. Systemic complications were also reported. All treatment side effects or adverse events were recorded. Data were retrospectively collected preoperatively and postoperatively at day 1, month 1 (30 ± 10 days after surgery), month 12 (360 ± 10 days after surgery), and 48 months (1440 ± 10 days) after surgery.

This study was conducted in agreement with the principles of the Declaration of Helsinki and was approved by the Local Ethics Committee, Reggio Emilia, Italy (protocol n. 2016/0024410). Informed written consent was obtained from both parents of all patients.

## 3. Statistical Methods

Continuous variables were reported as the means and standard deviations (SDs) or median and interquartile range (IQR) and categorical variables as frequencies and percentages.

The chi-square test was used to evaluate associations between categorical variables and patient groups. Student’s *t*-test or Mann–Whitney were used to assess differences in continuous variables. The Spearman correlation coefficient was calculated to assess correlations between variables.

Paired *t*-test and ANOVA were used to compare CDVA in the 2 groups during follow-up.

Repeated measures to analyze the time and group effects was performed with a generalized linear model (GLM). The analyses were performed using SPSS v.26 (IBM SPSS Statistics, Rome, Italy); all tests were two-tailed, and a *p*-value ≤ 0.05 was considered statistically significant.

## 4. Results

### 4.1. Population

The study included 32 eyes of 24 children (seven males and 17 females) affected by JIA and a positive antinuclear antibody (ANA). The Non-Biological Group consisted of 10 eyes (seven children, six females) with a mean age at uveitis onset in the first eye of 5.6 ± 4.1 years (range 0–11) undergoing cataract surgery at a mean age of 7.8 ± 4.9 years (range 1–14) (Table 1). The Biological Group included 22 eyes (17 children, 11 females) with a mean age at uveitis onset in the first eye of 4.9 ± 2.6 years (range 2–11) and a mean age at surgery of 10.8 ± 4.1 years (range 4–20) (Table 1). The median time between uveitis onset and cataract surgery was longer in the Biological Group (82.0 months, IQR 1–131) than in the Non-Biological Group (13 months, IQR 4–19). All patients presented controlled ocular inflammation for at least 3 months prior to surgery.

### 4.2. Medical Treatment

In the Non-Biological Group, all patients were treated with oral antimetabolites—four out of seven patients (57.1%) with MTX and three out of seven patients (42.9%) with AZA. Three patients underwent bilateral cataract surgery: two were in therapy with AZA and one with MTX. The median time of treatment with oral antimetabolites before surgery in the patients was 7 months (range 2–36) before surgery. All patients received additional perioperative intravenous steroids (10 mg/kg of methylprednisolone, starting two hours before surgery and for 3 days consecutively). The systemic treatment with antimetabolites lasted for a median of 17 months (range 4–36) after surgery.

In the Biological Group, the patients were treated with MTX or AZA for a median of 14 months (range 3–117) before surgery. Twelve out of 17 patients (70.6%) were additionally treated with ADA, 3/17 patients (17.6%) with IFX, and 2/17 patients with TCZ (11.8%). Five patients underwent bilateral cataract surgery: four were in treatment with ADA and one with IFX.

The systemic treatment with biologicals lasted for a median of 36.5 months (range 1–67) after surgery. All patients continued systemic therapy for at least one year postoperatively.

During follow-up, one patient developed a mild systemic allergic reaction to IFX and was shifted to ADA, while another developed anti-ADA antibodies and was therefore shifted to TCZ. No severe adverse events related to the systemic treatment were recorded.

### 4.3. Surgical Intervention

An IOL was implanted in the bag with posterior capsulorhexis, anterior vitrectomy, and optic capture in five eyes in the Non-Biological Group and in three eyes in the Biological Group. An IOL was implanted in the capsular bag with basal iridectomy and was postoperatively followed by Nd:YAG laser posterior capsulotomy in five eyes in the Non-Biological Group and in 19 eyes in the Biological Group. Band keratopathy was removed with EDTA chelation in one eye (10%) in the Non-Biological Group and in six eyes (27%) of the Biological Group.

### 4.4. Visual Outcomes

In the Non-Biological Group, the mean CDVA changed from 20/310 (1.19 ± 0.72 logMAR) preoperatively to 20/120 (0.78 ± 0.59 logMAR) at 12 months (*p* = 0.19) and to 20/191 (0.98 ± 0.97) at 48 months of follow-up (*p* = 0.45). In the Biological Group, the mean CDVA significantly improved, from 20/710 (1.55 ± 0.91 logMAR) preoperatively to 20/40 (0.30 ± 0.29 logMAR) at 12 months (*p* < 0.0001) and to 20/74 (0.57 ± 0.83 logMAR) at 48 months of follow-up (*p =* 0.001) (Figure 1).

In the GLM repeated measure model for CDVA, both groups improved significantly over the first 12 months, but the differences between the groups were significant only at 12 months (*p* = 0.006) (Figure 2). The ANOVA showed that the CDVA changes in the Biological Group were significant at all time points, while, in the Non-Biological Group, an improvement was significant only at one month after surgery (Figure 3).

### 4.5. Postoperative Complications

Postsurgical synechiae developed in 9/10 eyes (90%) in the Non-Biological Group compared to 15/22 in the Biological Group (68.2%) (*p* = 0.19). In the Non-Biological Group, three eyes (30%) developed cyclitic membranes: one was treated with Nd:YAG laser, while two required surgical removal. In the Biological Group, four eyes (18.2%) developed cyclitic membranes and received anterior vitrectomy. An IOL was explanted in three eyes of the Non-Biological Group (30%) and in three eyes of the Biological Group (13.6%) (*p* = 0.27). Postoperative ME developed in 1/10 eyes (10%) in the Non-Biological Group and in 7/22 eyes (31.8%) in the Biological Group (*p* = 0.19). The IOP increased over 21 mmHg in 3/10 eyes (30%) in the Non-Biological Group and in 3/22 eyes (13.6%) in the Biological Group. Filtering surgery was necessary in two eyes (20%) in the Non-Biological Group and in all three eyes (13.6%) with ocular hypertension in the Biological Group (*p* = 0.27). One patient in the Biological Group developed an inoperable closed funnel retinal detachment. Hypotonia was recorded only in the Non-Biological Group in 2/10 eyes (20%) (Table 2). No other surgical complications were reported.

## 5. Discussion

Cataract surgery with IOL implantation in the children with JIA-associated chronic anterior uveitis is controversial due to the high rate of complications [9]. Most of the studies available are retrospective, with small groups of patients [1,3,8,9,15,16,17,18,19,20,21,22]. The most important prognostic factors identified so far are intraocular inflammation control and patient selection [4,24].

In 1996, Probst and Holland were the first to report on IOL implants in patients with JIA uveitis. In their study, only corticosteroids were used to control inflammation, and postoperative complications resulted more commonly in younger patients, suggesting that IOL implantation may lead to severe complications [17].

Similarly, BenEzra and Cohen did not use any additional immunosuppressors and found that younger patients developed more severe and sustained complications [25]. Further studies have shown how an optimal intraocular inflammation control may lead to improved visual outcomes after cataract surgery in these patients [21]. Despite the evidence from several studies that cataract surgery in children’s eyes with uveitis is beneficial, IOL implantation is still debated [24,26]. Children with JIA uveitis can sometimes be poor candidates for contact lens, because they are often on long-term topical steroid drops, thereby increasing the chances of developing infective keratitis. Moreover, the presence of band keratopathy can make contact lens fitting difficult [4]. BenEzra and Cohen highlighted how contact lenses were poorly tolerated by children with unilateral aphakia, and IOL implants seemed preferable [25]. On the other hand, some authors chose to perform cataract surgery using the combined lensectomy–vitrectomy technique, with no IOL implant, and preferred the use of contact lenses [6,7,27,28,29].

Nowadays, cataract surgery with IOL implantation may be considered in well-selected children affected by JIA-associated uveitis, providing adequate immunosuppression with zero tolerance of inflammation [22,24,30].

This study compared the outcomes after cataract surgery with IOL implantation in patients affected by JIA-associated chronic anterior uveitis treated with two different pharmacological regimens. In our cohort, postoperative mean CDVA significantly improved in patients treated with biological and immunosuppressive drugs compared to those receiving immunosuppressor and systemic corticosteroids one year after cataract surgery (*p* < 0.001) and resulted in a more stable outcome at 48 months of follow-up (*p* = 0.001). This suggests a higher and more stable visual gain in the Biological Group compared to the Non-Biological Group.

The median time between uveitis onset and cataract surgery in our study was longer in the Biological Group (83.4 months, IQR 20.7–130.6) than in the Non-Biological Group (10.8 months, IQR 3.6–54.9) (*p* = 0.03). Long-term corticosteroid use could be one of the reasons associated with a shorter time frame between uveitis onset and cataract development in the Non-Biological Group [31].

The postoperative complications, including synechiae, cyclitic membrane, IOL explantation, postoperative glaucoma, and ME, were not statistically different between the two groups (Table 2). PCO was treated with Nd:YAG laser capsulotomies performed under topical anesthesia on all patients in the Biological Group.

The mean time between uveitis onset and cataract surgery was longer in the Biological Group (*p* < 0.005), making it possible to perform both surgery at an older age and Nd: YAG laser capsulotomies under topical anesthesia.

Our study has several limitations, including its retrospective design, the small sample size and relatively short follow-up, the lack of standardization of cataract grading, and the nonhomogeneous surgical treatment. Therefore, further clinical studies with larger sample sizes are warranted to highlight the influence of various systemic regimes on cataract development and surgical outcomes.

In conclusion, the immunosuppressive treatment based on biological drugs combined with antimetabolites in children affected by JIA-associated anterior uveitis undergoing cataract surgery can improve the visual outcome compared to nonbiological treatments, but it does not significantly reduce the postoperative ocular complications.

## Figures and Tables

**Figure 1 jcm-10-02437-f001:**
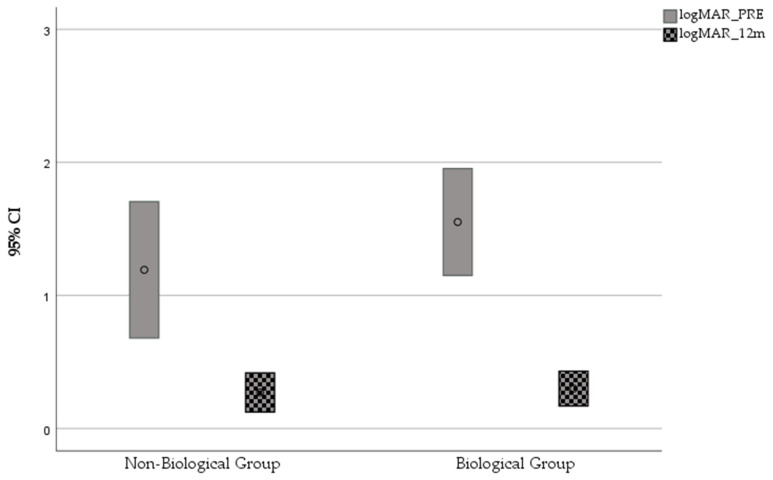
Mean CDVA and 95% CI in the 2 groups before and 12 months after surgery.

**Figure 2 jcm-10-02437-f002:**
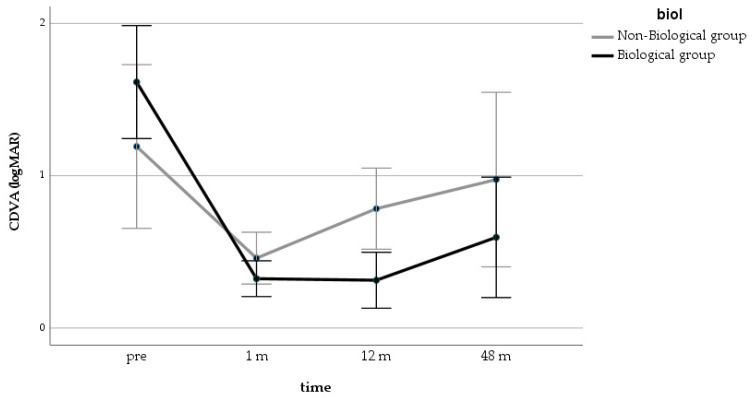
GLM repeated measure for CDVA in the 2 groups (estimated marginal means with 95% CI). GLM = generalized linear model; CDVA = corrected distance visual acuity.

**Figure 3 jcm-10-02437-f003:**
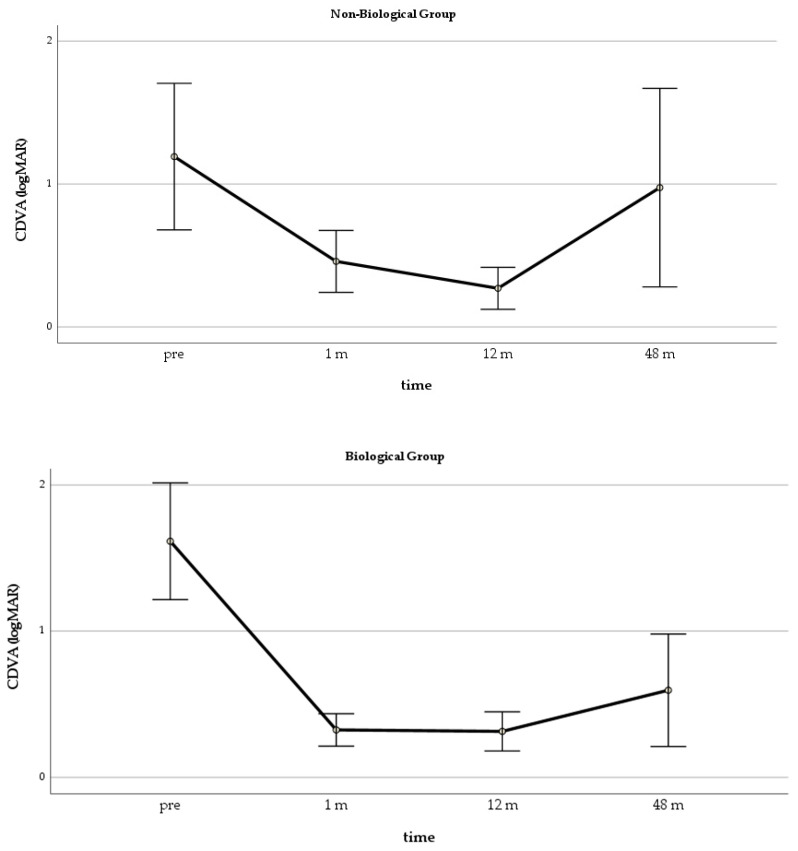
Mean CDVA changes in the 2 groups during follow-up. CDVA = corrected distance visual acuity.

**Table 1 jcm-10-02437-t001:** Baseline characteristics of the patients according to group.

		Non-Biological Group (*n* = 10 eyes)		Biological Group (*n* = 22 eyes)		Total (*n* = 32 eyes)	
		Mean ± SD	*n* (%)	Mean ± SD	*n* (%)	Mean ± SD	*n* (%)
Age at surgery, y		7.8 ± 4.9		10.8 ± 4.1		9.8 ± 4.5	
Age at diagnosis, y		5.6 ± 4.1		4.9 ± 2.6		5.1 ± 3.1	
Sex	m		1 (10)		9 (41)		10 (31)
	f		9 (90)		13 (59)		22 (69)
Eye involvement	bilateral		9 (90)		19 (86)		28(88)
	monolateral		1 (10)		3 (14)		4 (12)
pre-operative CDVA Snellen (LogMar)		20/310 (1.19 ± 0.72)		20/710 (1.55 ± 0.91)		20/551 (1.44 ± 0.86)	

SD = standard deviation; CDVA = corrected distance visual acuity.

**Table 2 jcm-10-02437-t002:** Postoperative complications compared between the two groups.

	Non-Biological Group (*n* = 10)	Biological Group (*n* = 22)	*p*-Value
Synechiae	9/10 (90%)	15/22 (68.2%)	0.19
Cyclitic membrane	3/10 (30%)	4/22 (18.2%)	0.45
IOL explantation	3/10 (30%)	4/22 (13.6%)	0.27
ME	1/10 (10%)	7/22 (31.8%)	0.19
Time ME (median), months	7	8	
IOP >21 mmHg	3/10 (30%)	3/22 (13.6%)	0.31
Filtering surgery	2/10 (20%)	3/22 (13.6%)	0.27
Hypotonia	2/10 (20%)	0	
Retinal detachment	0	1/22(0.04%)	

IOL = intraocular lens; ME = macular edema; IOP = intraocular pressure.

## Data Availability

The data presented in this study are available on request from the corresponding author.

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
