# Peer review of "Cataract Surgery with Intraocular Lens Implantation in Juvenile Idiopathic Arthritis-Associated Uveitis: Outcomes in the Era of Biological Therapy"

_jcm, 2021, doi:10.3390/jcm10112437_

Round 1
Reviewer 1 Report
The results of implanting IOLs in the eyes of 32 children with JIA are described. The authors report better visual outcomes in the group of patients who received biologics and antimetabolites vs oral steroids and antimetabolites. There are serious complications in both groups necessitating IOL explantation in 7 eyes, retinal detachment and hypotonia.
1. I don't think the discussion of contact lens wear for patients who are left aphakic is balanced. My experience is that these patients do well wearing contact lenses even though they often have band keratopathy. I have not had patients have the serious complications described in this report iwhen I left these eyes aphakic. I would recommend expanding the paragraph in the Discussion discussing the correction of these eyes when left aphakic with contact lenses and citing articles that report favorable results in aphakic eyes corrected with contact lenses.
Author Response
Point 2. Cataract in patients with JIA is a relatively uncommon disease, this has been also outlined in a review that states: “most of the studies conducted so far on this subject are retrospective, with small groups of patients.” (Phatak S, Lowder C, Pavesio C. Controversies in intraocular lens implantation in pediatric uveitis. J Ophthalmic Inflamm Infect. 2016). As stated at the prompt of the Methods section, this was a retrospective study. Sample size was not calculated since it was not a prospective study.
Points 3, 5 and 6. Thank you for the interesting comment. Following your suggestion, we changed the definition of BCVA with CDVA and added 20/20 values to the manuscript. In terms of adapting the structure to the refractive journals frame, it is our understanding that the Journal of clinical medicine has not mandatory guidelines to represent post-operative visual acuity outcome as far as the graphs correctly show the data. Moreover, our study is not solely related to the evaluation of post-operative visual outcome, but also describes post-operative complications and the potential of biological therapy in a complex clinical picture such as pediatric uveitis. In our setting the CDVA outcome parameter does not provide information relevant to the performance of the procedure from a refractive surgery point of view but rather indirectly supports better control of the underlying disease provided by biological therapy.
Point 4. The Introduction section of the manuscript has been reduced to three main paragraphs as suggested.
Reviewer 2 Report
The authors compare the outcomes of cataract surgery with intraocular lens (IOL) im-19 plantation in patients with juvenile idiopathic arthritis (JIA)-associated chronic anterior 20 uveitis treated with antimetabolite drugs and systemic corticosteroids (Non-Biological 21 Group) versus patients treated with antimetabolites and biological drugs (Biological 22 Group) and they found that immunosuppressive treatment with biological drugs can improve visual outcome after cataract surgery in patients with JIA-associated uveitis, but it does not significantly reduce postoperative ocular complications.
Major concerns
- I was a paper well written and well structured, but some point must be addressed before to continue publication process
- Sample size was small. Why? Sample size was calculated?
- visual and refractive outcomes must be described with journal of refractive surgery or journal of cataract and refractive methodology as the gold standard in visual and refractive outcomes. Find more info on: https://journals.healio.com/journal/jrs/submit-an-article
- reduce introduction into three main paragraph and provide only very concern information about the paper do not include extra o additional info with no relationship with the topic
- methods should be expanded with the preoperative assessment, surgical procedure, postoperative assessment..
- The paper should re-amended with an refractive surgery structure, please see any paper of journal of refractive surgery or journal of cataract of refractive surgery (Refractive and Visual Outcomes of SUPRACOR TENEO 317 LASIK for Presbyopia in Hyperopic Eyes: 24-Month Follow-up José-María Sánchez-González, Federico Alonso-Aliste, Jonatan Amián-Cordero, María Carmen Sánchez-González, Concepción De-Hita-Cantalejo PMID: 31498417 DOI: 10.3928/1081597X-20190815-01) and follow the same structured propose to all refractive surgery papers.
Author Response
Point 1. We thank the reviewer and agree that some authors prefer to leave the patients with JIA aphakic after cataract surgery and use contact lens. That approach may have clear advantages over IOL. A new sentence with proper references has been added to our Discussion (line 245).
Round 2
Reviewer 1 Report
My comments were addressed.
Author Response
Point 2. Cataract in patients with JIA is a relatively uncommon disease, this has been also outlined in a review that states: “most of the studies conducted so far on this subject are retrospective, with small groups of patients.” (Phatak S, Lowder C, Pavesio C. Controversies in intraocular lens implantation in pediatric uveitis. J Ophthalmic Inflamm Infect. 2016). As stated at the prompt of the Methods section, this was a retrospective study. Sample size was not calculated since it was not a prospective study.
Point 4. The Introduction section of the manuscript has been reduced to three main paragraphs as suggested.
Points 3, 5 and 6. Thank you for the interesting comment. Following your suggestion, we changed the definition of BCVA with CDVA and added 20/20 values to the manuscript. In terms of adapting the structure to the refractive journals frame, it is our understanding that the Journal of clinical medicine has not mandatory guidelines to represent post-operative visual acuity outcome as far as the graphs correctly show the data. Moreover, our study is not solely related to the evaluation of post-operative visual outcome, but also describes post-operative complications and the potential of biological therapy in a complex clinical picture such as pediatric uveitis. In our setting the CDVA outcome parameter does not provide information relevant to the performance of the procedure from a refractive surgery point of view but rather indirectly supports better control of the underlying disease provided by biological therapy.
Reviewer 2 Report
The authors have not responded at all to the proposed comments.Author Response
Point 1. We thank the reviewer and agree that some authors prefer to leave the patients with JIA aphakic after cataract surgery and use contact lens. That approach may have clear advantages over IOL. A new sentence with proper references has been added to our Discussion (line 245).